# Model-based inference of cell cycle dynamics captures alterations of the DNA replication programme

**Adolfo Alsina** [1,2]*, **Marco Fumasoni** [1], **Pablo Sartori** [1]

**1** Gulbenkian Institute for Molecular Medicine, Oeiras, Lisbon, Portugal, **2** GISC, Universidad Rey Juan Carlos, Móstoles, Madrid, Spain

☙ These authors contributed equally to this work.

* adolfo.alsina@urjc.es

## Abstract

The eukaryotic cell cycle comprises several processes that must be carefully orchestrated and completed in a timely manner. Alterations in cell cycle dynamics have been linked to the onset of various diseases, underscoring the need for quantitative methods to analyze cell cycle progression. Here we develop RepliFlow, a model-based approach to infer cell cycle dynamics from flow cytometry data of DNA content in asynchronous cell populations. We show that RepliFlow captures not only changes in the length of each cell cycle phase but also alterations in the underlying DNA replication dynamics. RepliFlow is species-agnostic and recapitulates results from more sophisticated analyses based on nucleotide incorporation. Finally, we propose a minimal DNA replication model that enables the derivation of microscopic observables from population-wide DNA content measurements. Our work presents a scalable framework for inferring cell cycle dynamics from flow cytometry data, enabling the characterization of replication programme alterations.

**Data availability statement:** RepliFlow is available at github.com/al-sina/repliflow.

## Author summary

The cell cycle is the coordinated sequence of events between two consecutive cell divisions. In eukaryotes, the cell cycle consists of four main phases: $G_1$, S, $G_2$, and mitosis. Understanding the constrains regulating cell cycle progression requires quantifying the amount of time that cells spend in each cell cycle phase. While several methods have been developed to measure cell cycle progression, a typical approach is to use flow cytometry, a technology that allows to rapidly measure the DNA content of thousands of cells. Whereas conventional analysis of flow cytometry data frequently depends on heuristic approximations, we have developed RepliFlow, a model-based computational method to infer the amount of time allocated to each cell cycle phase from DNA content distributions. Applicable to different cell types, RepliFlow additionally captures

**Funding:** M.F. acknowledges the support of the Horizon 2020 Marie Sklodowska-Curie Actions (101030203) and an FCT fellowship (2023.09068.CEECIND, https://doi.org/10.54499/2023.09068.CEECIND/CP2854/CT0003). Work in the Genome Maintenance and Evolution lab was supported by FCT (2022.07846.PTDC), EMBO (5349-2023), HFSP (RGEC28/2023, https://doi.org/10.52044/HFSP.RGEC282023.pc.gr.168580), the Gulbenkian Foundation (FCG) and the Gulbenkian Institute for Molecular Medicine Foundation (GIMM) to M.F.. The funders had no role in study design, data collection and analysis, decision to publish, or preparation of the manuscript.

**Competing interests:** The authors have declared that no competing interests exist.

alterations to the DNA replication dynamics without requiring specialized experimental techniques. We demonstrate the applicability of Repliflow across different datasets, establishing it as a robust framework for quantitative analysis of cell cycle dynamics.

## Introduction

The cell cycle is the series of events between successive divisions, requiring precise coordination of molecular duplication to ensure viability. Disruptions in this process can impair division and lead to genetic instability and cell death [1–3].

Most eukaryotic cell cycles have a highly conserved structure, being composed of four major phases [4]. During the first phase $G_1$ (Gap 1), cells prepare for DNA replication. In the subsequent S phase (synthesis) the DNA is replicated. Finally, cells enter $G_2$ (Gap 2) phase, which paves the way for M (mitosis), when the cell physically divides. A deep understanding of the constraints regulating the cell cycle requires the development of methods that can accurately quantify the timing of its phases across a wide spectrum of conditions.

Microscopy-based fluorophore reporters have been developed to track cell cycle dynamics at the single-cell level in different organisms [5–7]. While powerful, these methods are limited in scalability due to genetic requirements and lengthy acquisition times. A compelling alternative is represented by the use of DNA content as a proxy for cell cycle progression. With a single dye intercalating the DNA, and without the need for genetic manipulations, the DNA content of thousands of single cells can be readily measured using flow cytometry [8,9]. Because DNA content changes in a stereotypical manner during the cell cycle, this method can, in principle, resolve the relative abundance of three out of the four main cell cycle phases ($G_1$, S, and $G_2$/M) across hundreds of conditions simultaneously.

However, current approaches to infer cell cycle dynamics from the information present in the distribution of DNA content across an asynchronous population (DNA profiles) are not grounded on biophysical models of cell cycle regulation [10–12]. Instead, they typically rely on heuristic definitions of the cell cycle phases, which limits their interpretability. To address this, we develop RepliFlow, a new framework to infer cell cycle dynamics from flow cytometry data based on a biophysical model of the cell cycle. RepliFlow is cell-type agnostic and depends only on a small number of interpretable parameters, encoding not only the time allocated to each cell cycle phase but also the presence, as well as the character, of alterations in DNA replication dynamics.

Using a combination of published datasets and targeted experimental perturbations of the cell cycle dynamics, we show that we can accurately infer parameters characterising the cell cycle dynamics in a large number of different conditions and cell types, requiring solely DNA content measurements. Furthermore, by introducing a microscopic model of the DNA replication dynamics, we also show that, for wild-type cells with unperturbed dynamics, this approach can infer microscopic observables within the constraints of flow cytometry data, such as the typical replication fork velocity, with an accuracy on par with modern sequencing technologies at a fraction of the cost and effort. Taken together, our framework provides a quantitative and scalable approach to study time allocation in growing cell populations, paving the way to future studies aimed at understanding the biophysical constraints underlying this process.

## Results

### Inference of cell cycle dynamics recapitulates cell cycle time allocation in *S. Cerevisiae*

Consider a population of asynchronously growing cells. As each cell is in a different phase of its cycle (Fig 1A), the population is characterised by the empirical distribution of DNA abundance across cells, which depends on the relative duration of each phase. Here, we develop a general likelihood-based approach to infer the fraction of time allocated to each cell cycle phase from such distribution (see also Appendix A in S1 File for a detailed introduction to RepliFlow).

The centerpiece of our analysis is $\boldsymbol{y} = \{y_1, \dots, y_N\}$, the empirical distribution of DNA content in a population of $N$ cells. This distribution depends on both the cell cycle dynamics of individual cells and the age structure of the population. We can express the likelihood of observing the empirical DNA content distribution $\boldsymbol{y} = \{y_1, \dots, y_N\}$ as

$$\mathcal{L}(\boldsymbol{y}|\boldsymbol{\theta}) = \prod_{i=1}^{N} \int_0^1 P(t) f(y_i|\boldsymbol{\theta}, t) \mathrm{d}t, \qquad (1)$$

where $P(t)$ represents the cell cycle age distribution across the population, $t$ represents cell age, i.e. the time elapsed since the last division event; and $f(y|\boldsymbol{\theta}, t)$ is a function that quantifies the amount of DNA content in a cell of age $t$ and is characterised by parameters $\boldsymbol{\theta}$. The vector $\boldsymbol{\theta}$ includes parameters such as the duration of the different cell cycle phases and also parameters that account for the effect of technical noise. For a given DNA content distribution $\boldsymbol{y}$, we

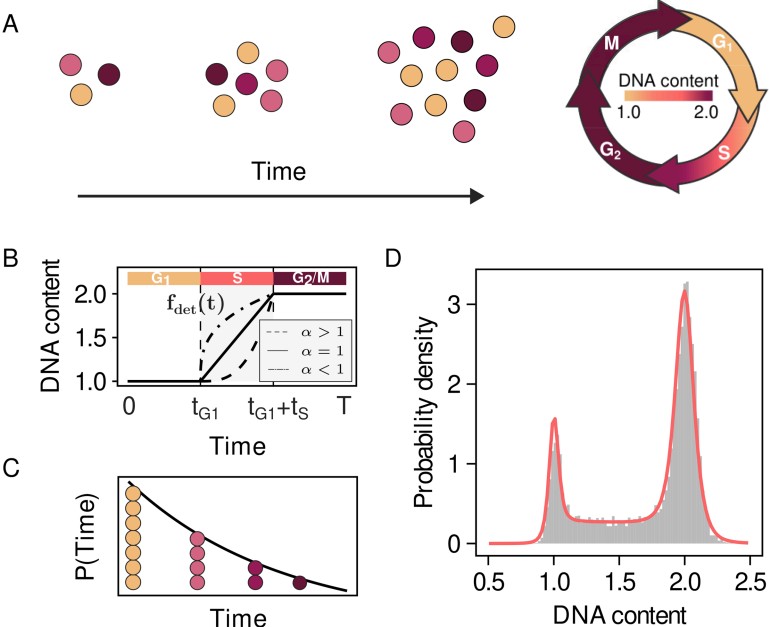

**Fig 1. Distribution of DNA content as a combination of single-cell and population level processes.** A) Schematic of the time evolution of an exponentially growing asynchronous cell population. B) Dynamics of DNA content during the length of the cell cycle. C) Distribution of cell ages, i.e. time elapsed since the last division event, in an exponentially growing asynchronous cell population. D) Distribution of DNA content in an asynchronous cell population and resulting maximum likelihood fit of our model (red).

infer the parameter values that maximise the log-likelihood function

$$\theta^* = \mathrm{argmax}_\theta \log \mathcal{L}(y|\theta) \,. \tag{2}$$

In other words, the parameter set that maximizes the likelihood, $\theta^*$, serves as our quantitative characterization of the cell cycle dynamics, capturing phase durations and other key features directly from the observed DNA content distribution.

To completely determine the model, it remains to specify the forms of $P(t)$ and $f(y|\theta, t)$. These contributions are schematically represented in Fig 1B and 1C. First, we consider the measured amounts of DNA to result from the combination of deterministic single-cell dynamics and technical noise (see Appendix A in S1 File). We represent the deterministic dynamics of the amount of DNA content (Fig 1B) by a continuous function

$$f_{\mathrm{det}}(t) = \begin{cases} 1 & 0 < t < t_{\mathrm{G_1}} \\ f_{\mathrm{det,S}}(t) & t_{\mathrm{G_1}} < t < t_{\mathrm{G_1}} + t_{\mathrm{S}} \\ 2 & t_{\mathrm{G_1}} + t_{\mathrm{S}} < t < 1 \end{cases}, \tag{3}$$

where $t_i$ is the fraction of time allocated to phase $i$ and $f_{\mathrm{det,S}}$ is a monotonically increasing function describing the change in DNA content during S phase. To account for technical noise, we model $f(y|\theta, t)$ as a probability distribution with mean $\mu = f_{\mathrm{det}}(t)$ and variance $\sigma^2$.

We model DNA replication dynamics during S phase using a minimal phenomenological model. While unperturbed replication kinetics are accurately described assuming constant DNA replication speed, to account for replication defects we introduce a single-parameter model encompassing both regular and perturbed replication dynamics:

$$f_{\mathrm{det,S}}(t) = 1 + \left( \frac{t - t_{\mathrm{G_1}}}{t_{\mathrm{S}}} \right)^\alpha \,. \tag{4}$$

For $\alpha = 1$ the model reduces to a linear model of DNA replication, whereas $\alpha > 1$ and $\alpha < 1$ correspond, respectively, to defects in early and late replication (Fig 1B).

Next, we focus on the cell cycle age distribution $P(t)$ (Fig 1C). In an exponentially growing population, it is well known that cell ages are distributed exponentially [13] (see Appendix A in S1 File).

Eq 1 together with the definitions of $f(y|\theta, t)$ and $P(t)$ constitute a closed optimization problem on the parameters $\theta$. Given an empirical distribution of DNA content, RepliFlow infers the allocated time fractions and technical noise parameters that maximise the likelihood function (see Appendix A in S1 File).

To validate our model, we tested it on the DNA profile of an exponentially growing asynchronous *S. cerevisiae* population (Fig 1D), correctly reproducing the main features of the data. For the DNA distribution of Fig 1D, obtained from a population grown in media supplemented with 2% glucose at $30°C$, RepliFlow estimates that cells allocate 9.7% ([8.99,10.34] 95% credible interval, see Appendix B in S1 File for a robustness analysis of the maximum likelihood estimate) of time to $G_1$ phase, 23.3% ([22.25,24.35] 95% credible interval) of time to S phase and 67% ([65.97,68.14] 95% credible interval) of time to $G_2$/M. Given the measured doubling time in these conditions of 117.2 min, these fractions correspond to absolute durations of 11.37 min, 27.30 min and 78.52 respectively. The inferred phase durations qualitatively align with the expected time spent in each phase under these conditions. Regarding replication dynamics, for this profile we find a value of $\alpha = 0.96$ ([0.90,1.04] 95% credible

interval), indicating that DNA replication proceeds at approximately constant speed throughout S phase, consistent with a linear increase in DNA content. RepliFlow is computationally efficient, requiring between 10 and 180 seconds on a standard laptop computer depending on the complexity and features of the DNA profile under consideration. Furthermore, benchmarking against common alternative frameworks [10–12] demonstrates that RepliFlow not only matches but often outperforms them, particularly when evaluating performance across all cell cycle phases simultaneously (Appendix C in S1 File).

Thus, our approach provides a new way of estimating cell cycle dynamics that is fast, robust and accurate.

## Analysis of the haploid yeast deletion collection captures alterations in the cell cycle

Having established our inference framework, we set out to show that it can accurately capture variability in the time allocated to the different cell cycle phases. With that goal in mind, we analysed flow cytometry data of the haploid yeast deletion collection, a comprehensive dataset of all non-essential single-gene deletion strains in haploid *S. cerevisiae* [14,15]. As several mutants impact key cell cycle processes, the deletion collection provides a suitable testing ground to assess how RepliFlow captures changes in the allocation of cell cycle time in budding yeast mutants.

The analysis of the deletion collection reveals large variablity in the relative time allocated to all three distinguishable phases across mutants (Fig 2A). Model fits were highly accurate for the majority of strains, with 95% achieving a fitting score $s>0.95$ (Fig 2B, see Appendix A in S1 File for details). Furthermore, we find a strong negative correlation between the percentage of time spent in $G_1$ and $G_2/M$ phases, as previously reported [16]. This correlation accounts for most of the variation between individual mutants, indicating that frequently an extension of one of the gap phases comes at the expense of the length of the other (S1 Fig). As the original experimental design is susceptible to plate-to-plate variation, in the following we normalise our results by the corresponding in-plate controls (z-scores, see Appendix D in S1 File for details).

The inferred parameter distributions (Fig 2C–2F) are sharply peaked around 0, indicating that the bulk of the deletion collection can be well described by the average time fractions allocated to the different phases across the whole collection together with a linear model of DNA replication (Fig 2F). Conversely, mutants in the the tails of the z-score distributions correspond to strains with shortened or extended phase durations, whereas the tails of the $\alpha$ distribution correspond to mutants with defects during early or late DNA replication, respectively. The homogeneity of the bulk of the deletion collection is further illustrated in Fig 2G, where most strains show only minor deviations in time allocation relative to the population average.

To assess whether our method reliably recovers known cell cycle defects, we examined representative mutants from the tails of these parameter distributions (highlighted in Fig 2A) for which the biological mechanism underlying the corresponding cell cycle alteration is well known. Several examples illustrate how RepliFlow identifies phase-specific alterations. For instance, deletion of the ribosomal subunit gene *RPL20B* results in a pronounced $G_1$ extension (Fig 2H), consistent with the reduced protein synthesis delaying attainment of the critical size required for S-phase entry [17]. Similarly, *DUN1* mutants, lacking a kinase that regulates dNTP production, exhibit prolonged S phase (Fig 2I) as genome replication slows [18–20]. In contrast, deletion of *HSL1*, a gene involved in cytokinesis, results in extended $G_2/M$ (Fig 2J) [21].

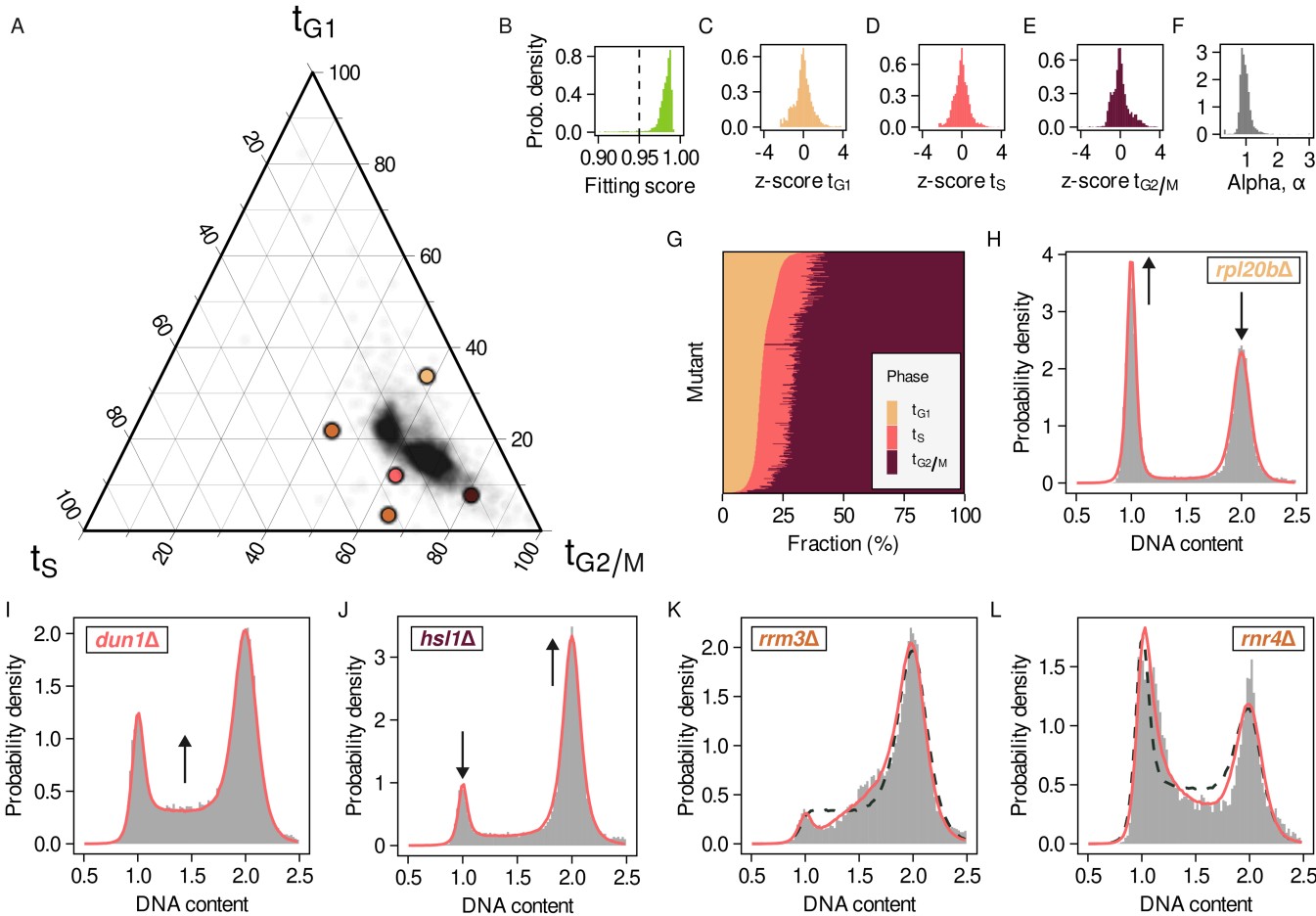

**Fig 2. Analysis of the yeast deletion collection recovers extended cell cycle phase durations and alterations of the replication programme.** A) Ternary plot of the fraction of time spent in each cell cycle phase. Colored points correspond to the profiles represented in panels H-L. B) Distribution of fitting scores of the deletion collection profiles. The dashed line corresponds to the minimum fitting score required for all subsequent analysis. C-E) Distribution of z-scores of the fraction of time spent in $G_1$ (C), S (D) and $G_2$/M (E) phase. F) Distribution of alpha values of the deletion collection profiles. G) Heatmap of the fraction of time allocated to $G_1$, S and $G_2$/M for all deletion collection mutants. H-J) Example profiles and fits of mutants with extended $G_1$ (H), S (I) and $G_2$/M phase (K). K-L) Example profiles and fits of mutants with fitted $\alpha$ values smaller (K) and larger (L) than one. Dashed lines correspond to a linear DNA replication model.

Taken together, these examples show that RepliFlow systematically captures variability in the time allocated to each cell cycle phase across mutants, recapitulating previously known cell cycle alterations.

Beyond phase lengthening, RepliFlow also detects alterations in replication dynamics through the parameter $\alpha$. As expected, mutants from the tails of the $\alpha$ distribution reveal distinct defects in early versus late DNA replication. Fig 2K shows the DNA profile of a *rrm3Δ* mutant strain. *RRM3* encodes for a DNA helicase, crucial for the removal of protein obstacles to the DNA replication forks [22–24]. In *rrm3Δ* mutants stalled forks delay the completion of DNA replication, leading to the observed build-up of cells with incompletely duplicated genomes at the end of S phase. We infer a value of the parameter $\alpha = 0.48 < 1$ (solid line, dashed corresponds to a linear model), reflecting the observed accumulation of cells close to the $G_2$/M peak and corresponding to the presence of defects during late DNA replication. The difference between the alpha and linear model is also reflected in the inferred time fractions

that are, respectively, $t_{G_1}$ = 3.43%, $t_S$ = 31.69% and $t_{G_2/M}$ = 64.88% for the alpha model, and $t_{G_1}$ = 1.24%, $t_S$ = 24.95% and $t_{G_2/M}$ = 73.81% for the linear version.

Conversely, *rnr4Δ* mutants, defective in dNTP synthesis [25,26], show difficulties initiating replication (Fig 2L). The presence of defects during early replication is reflected in the inferred value of $\alpha$ = 1.68. In this case the time fractions inferred with the alpha model are $t_{G_1}$ = 21.85%, $t_S$ = 34.83% and $t_{G_2/M}$ = 43.31%, in comparison with $t_{G_1}$ = 28.10%, $t_S$ = 30.92% and $t_{G_2/M}$ = 40.98% for a linear model.

In summary, RepliFlow not only quantifies shifts in the duration of individual cell cycle phases but also captures alterations in the replication program itself. These alterations are characterised using a minimal, uniparametric model of DNA content changes along the length of the cell cycle. Together, the proposed approach permits the identification of strains associated with specific cell cycle alterations from the tails of the parameter distributions.

## Quantifying graded perturbations of the cell cycle dynamics

Gene deletions are powerful tools to perturb cellular processes. However, deletions are restricted to non-essential genes and limited by the all-or-nothing nature of the perturbations they produce. To study the response to a graded perturbation, we exposed cells to methyl methanesulfonate (MMS), an alkylating agent that modifies bases and perturbs DNA replication [27,28]. To quantify the impact of MMS on the DNA replication dynamics, we systematically vary the concentration of the chemical across more than two orders of magnitude.

We find that for small concentrations of MMS, the DNA profile of MMS-treated cells resembles that of untreated cells (S2 Fig). However, increasing the MMS concentration causes the profiles to become increasingly skewed (Fig 3A), with an increase in the number of cells close to the $G_2/M$ peak, as a result of cells stalling before completing DNA replication.

This effect is captured by the replication parameter $\alpha$ (Fig 3B). As the MMS concentration is increased, we observe a decrease in the inferred values of $\alpha$, corresponding to a shift from a linear DNA replication programme ($\alpha \approx 1$) at low doses to altered replication dynamics at high doses. The DNA replication dynamics at high MMS concentrations are dominated by the presence of late defects difficulting completion, as reflected in the inferred values of $\alpha < 1$.

By independently measuring the doubling time of cells in each condition (Appendix D in S1 File), we quantify how the time allocated to each cell cycle phase changes as a function of the MMS concentration (Fig 3C). The total amount of time spent in each cell cycle phase is defined as $T_i = T_d t_i$, where $i = \{G_1, S, G_2/M\}$, $T_d$ is the doubling time of cells in these conditions and $t_i$ is the fraction of time spent in phase $i$. To test whether the time allocated to each cell cycle phase increases with MMS concentration, we performed two-sided t-tests comparing phase durations between the two lowest and the two highest MMS concentrations. We find that for large MMS concentrations the duration of S phase ($p = 7.52 \cdot 10^{-4}$) and $G_2/M$ ($p = 3.10 \cdot 10^{-4}$) is increased, whereas the change in the duration of $G_1$ is not statistically significant ($p = 0.3077$).

Our results are compatible with an interpretation where for low MMS concentrations the induced damage can be efficiently overcome, not impacting the overall cell cycle dynamics. However, the damage induced by larger concentrations of MMS requires cell to spend more time in S phase in order to be repaired and DNA replication to be completed. The transition between these two regimes can be readily identified thanks to the quantitative character of this approach.

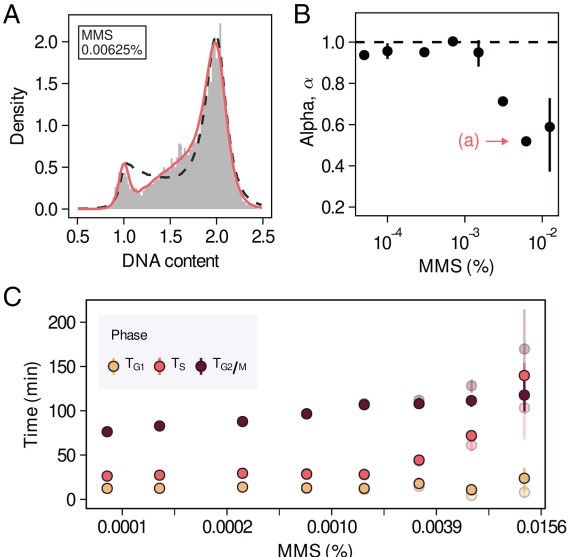

**Fig 3. Quantifying a graded perturbation of the DNA replication dynamics.** A) DNA profile of cells grown in the presence of MMS (0.00625%) presenting defects in late DNA replication. Solid lines correspond to fits of the $\alpha$ model to the data. Dashed lines correspond to fits of a linear model. B) Fitting parameter $\alpha$ as a function of the MMS concentration in the media. Points correspond to the average across four independent experiments and bars correspond to 95% confidence intervals. The MMS concentration corresponding to the profile shown in A) is highlighted. C) Time allocated to each cell cycle phase as a function of the MMS concentration inferred using a linear (transparent) and alpha (solid) replication models. Errorbars correspond to 95% confidence intervals.

## Characterisation of mammalian replication dynamics from DNA content alone

Although in previous sections we have focused on budding yeast datasets, the model is based on the conserved structure of the eukaryotic cell cycle. To illustrate the broader applicability of RepliFlow, we next show that we can also characterise cell cycle dynamics in mammalian cell lines.

A common approach to studying DNA replication in mammalian cells involves nucleotide incorporation assays [29–31]. Fluorescent analogs such as 5-Ethynyl-2'-deoxyuridine (EdU) integrate into newly synthesized DNA (Fig 4A), with EdU signal intensity reflecting replication rate and the number of actively replicating cells. Commonly, the fraction of cells in each cell cycle phase is calculated using a gating approach based on a geometric definition of the various cell cycle phases in the 2D DNA content-EdU plane (Fig 4B). While convenient, this approach is inherently ad hoc and can be difficult to apply consistently across conditions.

We applied our framework to the EdU incorporation experiments of [32], where the authors screened mutants with altered response to inhibitors of DNA replication initiation. Using RepliFlow, we inferred the fraction of time allocated to each cell cycle phase directly from DNA content and compared them against the estimates obtained from a conventional gating strategy (Appendix E in S1 File). In Fig 4C–4E we show that RepliFlow reliably estimates the fraction of time allocated to the three cell cycle phases under consideration from DNA content alone. The strong correlation between the estimates obtained by RepliFlow and the gating approach shows that we can indeed estimate the fraction of cells in different cell cycle phases from DNA content alone in a wide range of conditions, including pathological ones that strongly disturb cell cycle progression.

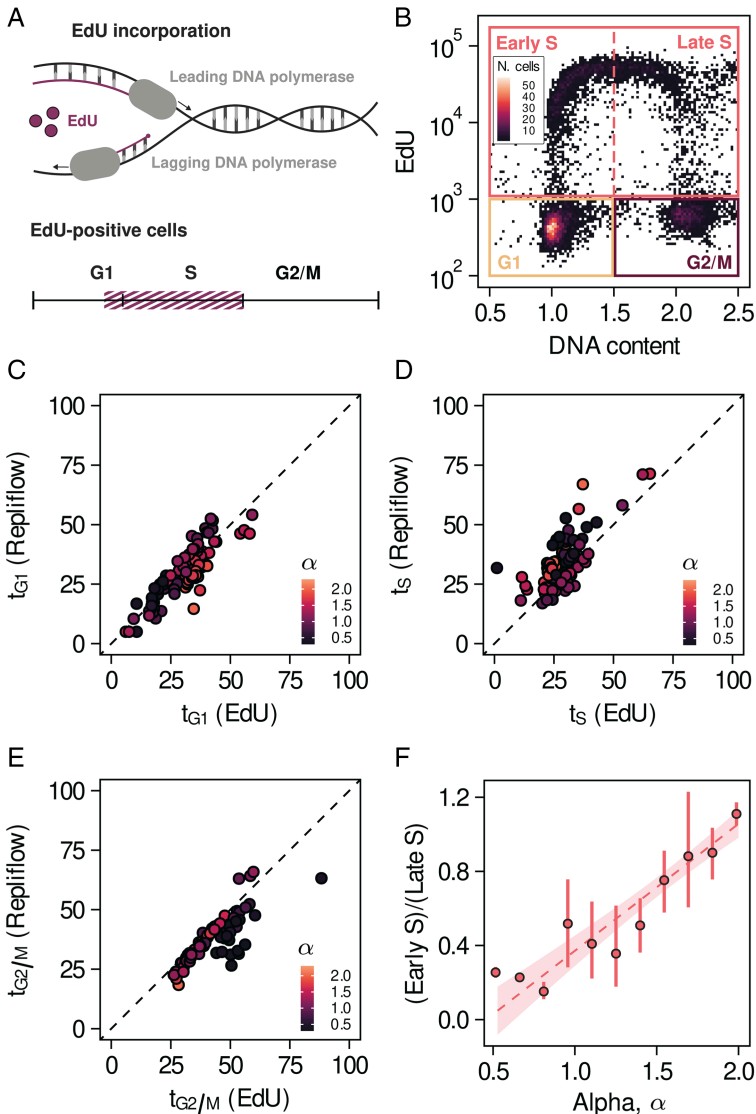

**Fig 4. Replication dynamics of mammalian cells obtained from DNA content alone.** A) Schematic of an EdU incorporation experiment. B) 2D histogram of EdU signal intensity and DNA content (data from Rainey et al., *Cell Reports*, 2020). Color corresponds to the number of events inside each bin. C) Fraction of time allocated to $G_1$ phase estimated from the EdU-DNA content histograms vs estimated from the DNA profiles using RepliFlow. D) Fraction of time allocated to S phase estimated from the EdU-DNA content histograms vs estimated from the DNA profiles using RepliFlow. E) Fraction of time allocated to $G_2/M$ phase estimated from the EdU-DNA content histograms vs estimated from the DNA profiles using RepliFlow. F) Ratio of the fraction of cells in early and late S phase estimated from the EdU-DNA content histograms as a function of the fitting parameter $\alpha$. The dashed line represents the trend of a linear regression, with the shaded region indicating the corresponding 95% confidence interval. Error bars correspond to bootstrapped 95% confidence intervals.

More importantly, beyond reproducing the fraction of time allocated to each phase, RepliFlow also reflects replication dynamics. The correspondence between RepliFlow and the gating approach is reflected in the correlation between the parameter $\alpha$ and the ratio of cells in early versus late S phase, as determined by EdU intensity and DNA content (Fig 4F). Early (late) S phase cells are characterised by an EdU signal intensity above $2 \cdot 10^3$ and DNA content

values below (above) 1.5 copies (Fig 4B). Our results show that $\alpha$ serves as a proxy for the type of DNA replication defects, as it accurately captures alterations in the replication programme, in agreement with the results of the gating approach. Thus, the proposed method offers a robust alternative to quantify replication dynamics from DNA content alone consistent with more costly and time consuming experimental approaches.

## Microscopic model of the replication dynamics

While our previous analyses relied on a phenomenological description of DNA replication, RepliFlow is flexible enough to incorporate mechanistic models of DNA content dynamics. To illustrate this modularity, we implemented a minimal microscopic model of DNA replication that does not take into account the spatial organisation of the genome. Although spatially explicit models have provided detailed theoretical descriptions of replication dynamics in eukaryotes [33–38], our simplified formulation nevertheless enables the extraction of quantitative microscopic information directly from DNA content profiles.

Fig 5A schematically illustrates the main components of the microscopic model. In brief, DNA replication starts from specific locations along the genome called origins of replication. During replication phase, licensed origins fire with rate $\gamma$. Upon firing, each origin recruits two replication forks that travel the genome in opposite directions (left) at speed $v$. When two replication forks meet, the DNA between their corresponding origins has been fully replicated and they detach from the DNA (center). As origin firing is a stochastic process, some origins are passively replicated and rendered inactive for the reminder of the cycle (right).

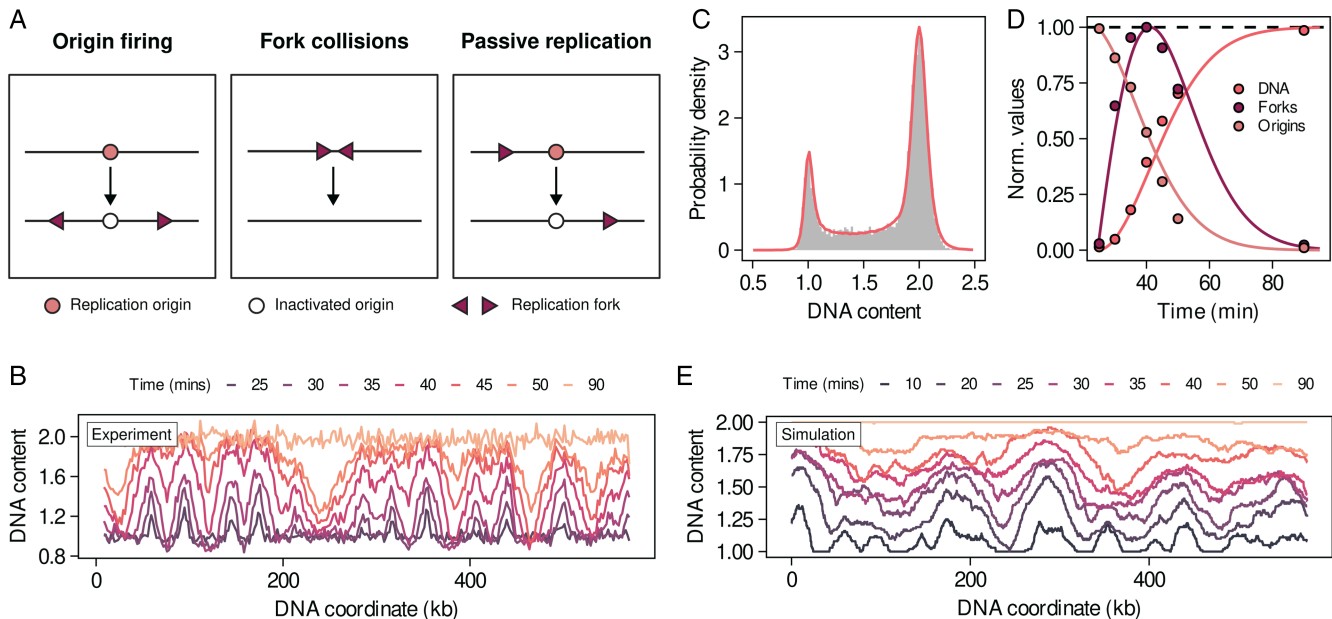

**Fig 5. A minimal model of DNA replication quantitatively reproduces microscopic replication dynamics.** A) Cartoon of the processes included in our phenomenological microscopic DNA replication model. B) Time series of DNA content across chromosome V for synchronised populations of budding yeast obtained with DNA sequencing. Data taken from Müller et al., *Nucleic acids research*, 2014. [39] C) DNA profile and fit of the data shown in Fig 1D using our microscopic DNA replication model. D) Comparison of the microscopic model predictions against data from Müller et al., *Nucleic acids research*, 2014. [39] The solid lines correspond to a simulation of the coarse-grained model calibrated using data from Fig 1D. Points correspond to data extracted from the deep sequencing time course (Fig 5B). E) Simulated dynamics of DNA content in chromosome V of budding yeast using the reported origin locations in this chromosome and the microscopic parameters inferred from C.

We propose a mean-field microscopic model of DNA replication describing the temporal evolution of the number of active forks $n$ over time, the number of licensed origins $\omega$ and the fraction of replicated DNA $\phi$

$$\frac{\mathrm{d}}{\mathrm{d}t}\phi = \nu n\,, \tag{5}$$

$$\frac{\mathrm{d}}{\mathrm{d}t}n = 2\gamma\omega - \nu\frac{n^2}{1-\phi}\,, \tag{6}$$

$$\frac{\mathrm{d}}{\mathrm{d}t}\omega = -\gamma\omega - \nu\frac{\omega n}{1-\phi}\,, \tag{7}$$

where $\nu$ is the fork velocity and $\gamma$ the origin firing rate. A detailed derivation of this model can be found in Appendix F in S1 File.

To test whether RepliFlow together with the microscopic model can accurately infer microscopic parameters from DNA content profiles we turn to the synchronous deep sequencing experiments from Müller et al. [39] (Fig 5B). The high temporal and spatial resolution of these experiments makes them a suitable baseline against which to test the performance of the model. We extract from the DNA sequencing data the time series of the relevant quantities for the microscopic model: the fraction of replicated DNA, the number of licensed origins and the number of active forks (see details in the Appendix F of S1 File).

We fitted the microscopic parameters directly from bulk DNA content measurements by replacing $f_{\mathrm{det,S}}$ in Eq 3 with $\phi$, and expanding the parameter set $\theta$ to include $(\nu, \gamma)$. In Fig 5C we show the fit to a WT DNA profile. We obtain parameter estimates $\nu = 1.76$ kb/min and $\gamma = 0.015$ min$^{-1}$, consistent with previous reports of replication fork velocity in budding yeast [40]. In addition, the inferred time fractions ($t_{\mathrm{G_1}} = 7.10\%$, $t_{\mathrm{S}} = 23.03\%$ and $t_{\mathrm{G_2/M}} = 69.87\%$) are consistent with those estimated using the phenomenological alpha model.

Using these estimates for the microscopic parameters, in Fig 5D we show that the microscopic model accurately reproduces the dynamics of the different observables extracted from the sequencing experiments of [39]. Importantly, in this comparison the microscopic model has no free parameters, as the fork velocity and origin firing rate are inferred from the DNA profile in Fig 5C. Therefore, fitting the microscopic model to flow cytometry data we can predict the dynamics of deep sequencing experiments. In Fig 5E we show a simulated time series of the DNA content distribution for chromosome V of budding yeast during synchronous replication (see Appendix F in S1 File for details). Our results reveal that some spatial features can be reproduced starting from the reported origin locations and the microscopic parameters inferred using the model.

In summary, we have shown that a minimal DNA replication model depending only on two parameters can provide quantitative insights into the microscopic workings of DNA replication at a fraction of the cost, effort and time of sophisticated sequencing technologies. While our microscopic model recapitulates quantitative features of unperturbed DNA replication, future RepliFlow-based extensions could be developed to capture altered replication programs.

## Discussion

The study of cell cycle dynamics has long benefited from a wealth of available experimental techniques that allow to measure cell cycle progression both at the molecular [41,42] and the population levels [27]. To extract maximum information from large datasets, new theoretical and computational tools that leverage our knowledge about cell cycle dynamics are

required. Here, we have introduced RepliFlow, a new method to infer cell cycle dynamics from flow cytometry data based on a biophysical model of DNA content changes during cell cycle progression.

RepliFlow extracts the duration of most cell cycle phases from DNA profiles of asynchronous cell populations. These profiles can be obtained at a small cost and in a short time using a conventional flow cytometer, making our method particularly suitable for high-throughput analysis. We characterise DNA profiles in terms of the combination of single-cell and population level contributions. At the single-cell level, we introduce a biophysical model of DNA content dynamics during the length of the cell cycle. Together with the age structure of the population and the effect of technical noise, these contributions characterise the experimentally measured DNA profiles.

Compared to common alternatives [10–12], RepliFlow uses fewer parameters, all of which are biologically interpretable, to infer the duration of the main cell cycle phases and the dynamics of DNA replication. Furthermore, the model captures alterations in the replication dynamics encoded in a single parameter $\alpha$, that permits us to distinguish defects in early and late DNA replication.

Using chemical perturbations of the cell cycle, we showed that a quantitative approach can identify the transition between different regimes in response to a graded perturbation. Together with the analysis of the yeast deletion collection, this highlights how the proposed framework serves as a quantitative tool for efficiently scanning different experimental conditions to detect specific defects. In combination with the development of custom-made microscopic models, this approach can provide information about how different perturbations (dosages) influence DNA replication at the molecular level.

Furthermore, we also showed that RepliFlow generalizes beyond yeast. When applied to mammalian EdU incorporation data, it recapitulated phase durations with high accuracy and revealed replication dynamics consistent with the results of more costly experimental assays. Likewise, by introducing a minimal microscopic replication model we showed that our framework, even without spatial structure, can recover fork velocity and origin firing rates in WT cells with unperturbed dynamics from DNA content alone, achieving good agreement with deep sequencing benchmarks. However, the current microscopic model is restricted to unperturbed DNA replication, and additional developments will be required for RepliFlow-based extensions to capture altered replication programs. Taken together, our work provides a robust and scalable quantitative approach to infer cell cycle dynamics from flow cytometry data.

However, currently RepliFlow relies on some assumptions that may limit its application in specific contexts. First, we have exclusively focused on populations undergoing steady exponential growth, where the age-distribution across the population is given by Equation A5. The analysis of populations in different regimes, would require quantifying the mode of growth alongside the cell cycle dynamics and deriving the corresponding age-distribution. Populations not undergoing steady growth, for which the replication programme is time dependent, fall outside the scope of our approach. Second, our framework does not currently account for cell-to-cell variability, so all inferred quantities represent population averages. Recent studies reveal that microscopic replication parameters can vary across individual cells [43], and that even sister cells do not progress synchronously through S phase [44]. While our model could incorporate variability (see Appendix A in S1 File), doing so would require additional parameters, increasing complexity and computational cost. Ad-hoc future extensions of RepliFlow can therefore be developed to accommodate specific usage in conditions where our general assumptions are not applicable.

Importantly, the modular nature of our framework can be exploited to develop custom-made models of DNA replication, opening the way to further quantitative investigation of the dynamics of the eukaryotic cell cycle in perturbed or pathological conditions. A promising direction for future work is the development microscopic models of DNA replication for specific perturbations that reproduce the observed DNA profiles where the underlying molecular details are not yet well understood. While the resolution of flow cytometry inherently limits the ability to distinguish between alternative mechanisms such as reduced origin licensing or delayed firing, RepliFlow can nevertheless extract meaningful quantitative information when guided by biological insight. Crucially, these limitations reflect the information content of flow cytometry data rather than shortcomings of the model itself. What RepliFlow uniquely provides is a systematic framework to detect perturbations and quantify their impact on DNA replication dynamics from DNA content alone, a capability that was not accessible with previous approaches. To obtain molecularly detailed insights, a biologically informed microscopic model is required, and RepliFlow offers the principled connection between such models and flow cytometry measurements.

Finally, the scalability of the proposed framework allows to quickly scan a large number of experimental conditions. Inspired by research on the constrains underlying proteome allocation [45–47], an important question for future work is whether there are any fundamental constraints that regulate the time allocated to the different cell cycle phases and how these depend on environmental variables.

## Supporting information

**S1 File. Appendices containing detailed mathematical derivations and benchmarking results against alternative methods**.
(PDF)

**S1 Fig. Paired plot depicting correlations between the inferred parameters for the yeast deletion collection**. Diagonal plots correspond to the marginal distribution of the corresponding parameters, whereas off-diagonal scatter plots show the correlation between each pair of model parameters.
(EPS)

**S2 Fig. DNA profile of cells grown in the presence of a low concentration of MMS** (0.0015%). The red solid line corresponds to the fit of the alpha model to the data. The black dashed line corresponds to the fit of a linear model.
(EPS)

**S3 Fig. Paired plot depicting correlations between 3200 parameter combinations sampled from the posterior parameter distribution obtained from the data in Fig 1D**. Diagonal plots correspond to the marginal distribution of the corresponding parameters, whereas off-diagonal scatter plots show the correlation between each pair of model parameters. The red dashed lines correspond to the maximum likelihood estimate of the parameters.
(EPS)

**S4 Fig. Distribution of DNA content for the profile shown in Fig 1D**. Solid black line represents the maximum likelihood fit. Red dashed lines correspond to profiles generated from the parameter combinations sampled from the posterior distribution represented in S4 Fig.
(EPS)

**S5 Fig. Synthetic profiles used as a benchmark for the comparison between the three methods under consideration: Repliflow, Dean-Jett and thresholding**. All profiles correspond to populations with 20% of cells in $G_1$ phase, 20% of cells in S phase and the remaining 60% in $G_2$/M. Dashed vertical lines correspond to the thresholds chosen for classification of cells into $G_1$, S and $G_2$/M using the thresholding approach. A) Distribution of DNA content of an asynchronous population with normal progression through S phase. B) Distribution of DNA content of an asynchronous population with late replication issues. C) Distribution of DNA content of an asynchronous population with early replication issues.
(EPS)

**S6 Fig. Relative error defined as $\Delta t_i = (t_i - t_{i,0})/t_{i,0}$, where $t_i$ is the time fraction allocated to phase i and $t_{i,0}$ is the ground truth for the three methods (Repliflow, Dean-Jett and thresholding) evaluated across the synthetic dataset and selected profiles from Rainey et al., Cell Reports, 2022**. Only those profiles from Rainey et al. for which both $G_1$ and $G_2$/M phase could be unambiguously identified were considered for this analysis.
(EPS)

## Acknowledgments

The authors want to thank David Soriano-Paños, Florian Pflug and Simone Pigolotti for comments on an earlier version of this manuscript, Michael D. Rainey and Corrado Santocanale for kindly sharing the dataset used in Fig 4 and the Flow Cytometry Platform of the Gulbenkian Institute for Molecular Medicine for their technical support. A. A. acknowledges the support of the IGC PONTE postdoctoral programme.

## Author contributions

**Conceptualization:** Adolfo Alsina, Marco Fumasoni, Pablo Sartori.

**Formal analysis:** Adolfo Alsina.

**Investigation:** Adolfo Alsina, Marco Fumasoni.

**Methodology:** Adolfo Alsina.

**Resources:** Marco Fumasoni.

**Software:** Adolfo Alsina.

**Supervision:** Marco Fumasoni, Pablo Sartori.

**Visualization:** Adolfo Alsina.

**Writing – original draft:** Adolfo Alsina.

**Writing – review & editing:** Adolfo Alsina, Marco Fumasoni, Pablo Sartori.

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
