## [Decision Letter · Decision Letter 0]

12 Jun 2025

PCOMPBIOL-D-25-00798

Model-based inference of cell cycle dynamics captures alterations of the DNA replication programme

PLOS Computational Biology

Dear Dr. Alsina,

Thank you for submitting your manuscript to PLOS Computational Biology. After careful consideration, we feel that it has merit but does not meet PLOS Computational Biology's publication criteria as it currently stands. Therefore, we invite you to submit a revised version of the manuscript that addresses the points raised during the review process. However, this does not guarantee a positive decision on the revised manuscript.

Please submit your revised manuscript within 60 days Aug 12 2025 11:59PM. If you will need more time than this to complete your revisions, please reply to this message or contact the journal office at ploscompbiol@plos.org. Please include the following items when submitting your revised manuscript:

We look forward to receiving your revised manuscript.

Kind regards,

Jing Chen

Academic Editor

PLOS Computational Biology

Mark Alber

Section Editor

PLOS Computational Biology

**Additional Editor Comments :**

The reviewers have mixed opinions about the general value of the work. One reviewer think it is generally suitable for the journal, while another criticized about the lack of valuable biology insights. Every reviewer, as well as myself, agrees on several major issues. First, the claim they about the performance of the tool compared to others was not supported by data. Second, some of the model assumptions are questionable. Third, lack of clarity makes judgement of the value of the work difficult. If the authors can successfully address these issues, the work will be considered for publication.

**Journal Requirements:**

At this stage, the following Authors/Authors require contributions: Adolfo Alsina, Marco Fumasoni, and Pablo Sartori. Please ensure that the full contributions of each author are acknowledged in the "Add/Edit/Remove Authors" section of our submission form.

4) Your manuscript is missing the following section headings: Abstract, and Introduction. Please ensure that the section heading levels are clearly indicated in the manuscript text, and limit sub-sections to 3 heading levels. An outline of the required sections can be consulted in our submission guidelines here:

5) Please upload all main figures as separate Figure files in .tif or .eps format. For more information about how to convert and format your figure files please see our guidelines: 

6) We notice that your supplementary Figures, and information are included in the manuscript file. Please remove them and upload them with the file type 'Supporting Information'. Please ensure that each Supporting Information file has a legend listed in the manuscript after the references list. Please ensure that the supplementary figures are labeled as "S1 Figure", "S2 Figure", and so forth.

7) When completing the data availability statement of the submission form, you indicated that you will make your data available on acceptance. We strongly recommend all authors decide on a data sharing plan before acceptance, as the process can be lengthy and hold up publication timelines. Please note that, though access restrictions are acceptable now, your entire data will need to be made freely accessible if your manuscript is accepted for publication. This policy applies to all data except where public deposition would breach compliance with the protocol approved by your research ethics board. If you are unable to adhere to our open data policy, please kindly revise your statement to explain your reasoning and we will seek the editor's input on an exemption. Please be assured that, once you have provided your new statement, the assessment of your exemption will not hold up the peer review process.

8) Please amend your detailed Financial Disclosure statement. This is published with the article. It must therefore be completed in full sentences and contain the exact wording you wish to be published.

3) If any authors received a salary from any of your funders, please state which authors and which funders.

9) Please ensure that the funders and grant numbers match between the Financial Disclosure field and the Funding Information tab in your submission form. Note that the funders must be provided in the same order in both places as well. Currently, the order of the grants is different in both places. "The Gulbenkian Foundation (FCG) and the Gulbenkian Institute for Molecular Medicine Foundation (GIMM)" are missing from the Funding Information tab.

**Reviewers' comments:**

Reviewer's Responses to Questions

**Comments to the Authors:**

**Please note that one of the reviews is uploaded as an attachment.**

Reviewer #1: This paper presents an interesting mathematical model of flow cytometry cell cycle profiles. It recapitulates previous findings pertaining to cell cycle perturbations of yeast mutants. While the approach can quantify cell cycle alterations including perturbations within S phase specifically, it is not clear that the approach is superior to other analyses of the same data. In that respect, the authors do not present a comparison and analysis of their data relative to other approaches for analyzing flow cytometry data.

The authors claim that they can infer perturbations of replication dynamics and also infer parameters of replication dynamics in general. I do not agree with these statements. While they can identify changes specifically in early vs late S phase, flow cytometry and their model cannot distinguish between replication origin loss/gain, changes in origin firing times, and changes in fork speed. It certainly cannot determine where in the genome any such changes (or the normal program for that matter) originate from. A delay in late S phase, for example (and as reported for RRM3 KO, similar to previous reports), could be due to loss of some origins, delays in a variety of replication origin subsets, or slowed replication forks, at specific locations or genome-wide.

Overall, while the mathematical models developed here may be of interest to researchers specializing in such approaches, the biological insights obtained with this work are limited.

Reviewer #2: In this manuscript the authors present a cell cycle based maximum likelihood based model to infer cell cycle distributions and replication dynamics using flow cytometry data solely stained with a DNA dye (i.e., DAPI). I think this approach is a very welcome addition to the cell cycle analysis toolkit. This is especially strong approach to analyze cell cycle distributions over a large set of perturbations, as demonstrated using the genetic knockout library in yeast. I think this manuscript is suitable for publication in PLoS Computational Biology, following addressing my concerns

Concerns

Although I appreciate the technical nature of this journal, I feel that the manuscript should be re-written with a larger audience in mind. Adoption of these methods is critical for their success. Therefore, I like to see two changes:

• The entire manuscript (especially the result section) should be extensively re-written to read less like a technical manual and take the time to explain design decisions for their application so that their target audience which I assume are both wet-lab biologists as well as computational biologist are able to digest this manuscript. The authors can describe the formulas and nitty-gritty details in the methods section of their manuscript.

• Code repositories should be made available with clear instructions on how to run the associated code (i.e., vignette). This might be even included in the manuscript as Supplementary documentation.

The assumption that even monogenic cell lines are progressing homogenous through the cell cycle is not necessarily true. For instance, using single cell sequencing revealed that cells at similar stages in S-phase can display different number of forks and replication speeds (van den Berg et al., Nature Methods, 2024). More recently, multigenerational time-lapse imaging has revealed that even sister cells are not progressing at similar rates through the cell cycle (Panagopoulos et al., Nature, 2025). I think the authors should at least cite these works and discuss their implications to their work. But I think a deeper exploration of variability parameters on their maximum likelihood estimation will increase the robustness of their method.

The authors state G1 are easily identifiable form S-phase cells based on the data from the Rainey et al., 2020 paper. However, I do not understand why they cannot use the G2 population from these data as this seems to be as easily identifiable in these data. It is very unclear why Fig. 8/ Suppl.Fig.3 the author opted to select a Cdc7i for a perturbation from Rainey et al., which I had to figure out myself by comparing resemblance between Suppl. Fig. 3 and Fig. 3a,c,j from Rainey et al., 2020. I personally do not see the problem in identifying the G2 population in these data. Therefore, I strongly feel the authors need to show that their model also predicts G2 proportions in a similar way they can predict fraction of G1 populations.

I am not exactly clear what the additional contribution of their microscopic model of replication dynamics is. In my mind, you still require the position of licensed origins, number of forks and position and fraction of replicated genome. Formally, the only thing this exercise adds is a speed and an origin firing rate? Do these speeds and orgin firing rates change in some of the yeast knockout strains? Are these in line with published literature? If the authors want to include this microscopic model in their revisions, they need to extend their analysis to other strains (i.e., knockouts) and show its applicability beyond a single condition.

The legends, especially of the Supplementary Figures, are absent or meager. The authors need to expand these to describe all features present in the Figures.

Reviewer #3: Attached

**Have the authors made all data and (if applicable) computational code underlying the findings in their manuscript fully available?**

Reviewer #1: None

Reviewer #2: **No: **the code is unavailable as far as I can tell

Reviewer #3: None

PLOS authors have the option to publish the peer review history of their article (what does this mean?). If published, this will include your full peer review and any attached files.

Reviewer #1: No

Reviewer #2: No

Reviewer #3: No

**Figure resubmission:**
---

## [Decision Letter · Decision Letter 1]

29 Sep 2025

Dear Dr Alsina,

We are pleased to inform you that your manuscript 'Model-based inference of cell cycle dynamics captures alterations of the DNA replication programme' has been provisionally accepted for publication in PLOS Computational Biology.

Best regards,

Jing Chen

Academic Editor

PLOS Computational Biology

Mark Alber

Section Editor

PLOS Computational Biology

Reviewer's Responses to Questions

**Comments to the Authors:**

Reviewer #1: The authors have addressed my concerns, thank you

Reviewer #2: The authors have adressed all my concerns and the alterations have made have drastically improved their work. I am in full favor of acceptence.

Reviewer #3: The authors respond to my comments successfully.

**Have the authors made all data and (if applicable) computational code underlying the findings in their manuscript fully available?**

Reviewer #1: Yes

Reviewer #2: Yes

Reviewer #3: Yes

PLOS authors have the option to publish the peer review history of their article (what does this mean?). If published, this will include your full peer review and any attached files.

Reviewer #1: No

Reviewer #2: **Yes: **Jeroen van den Berg

Reviewer #3: No

---

## [Editor Report · Acceptance letter]

PCOMPBIOL-D-25-00798R1

Model-based inference of cell cycle dynamics captures alterations of the DNA replication programme

Dear Dr Alsina,

I am pleased to inform you that your manuscript has been formally accepted for publication in PLOS Computational Biology. Your manuscript is now with our production department and you will be notified of the publication date in due course.

With kind regards,

Zsofia Freund
